# Unveiling Various Facades of *Tinospora cordifolia* Stem in Food: Medicinal and Nutraceutical Aspects

**DOI:** 10.3390/molecules28207073

**Published:** 2023-10-13

**Authors:** Varisha Anjum, Uday Bagale, Ammar Kadi, Irina Potoroko, Shirish H. Sonawane, Areefa Anjum

**Affiliations:** 1Department of Food and Biotechnology, South Ural State University, Chelyabinsk 454080, Russia; bagaleu@susu.ru (U.B.); ammarka89@gmail.com (A.K.); irina_potoroko@mail.ru (I.P.); 2Department of Chemical Engineering, National Institute of Technology, Warangal 506004, India; shirish@nitw.ac.in; 3Department of Ilmul Advia, School of Unani Medical Education and Research, Jamia Hamdard, New Delhi 110062, India; ariareefa@gmail.com

**Keywords:** *Tinospora cordifolia*, medicine, functional food, nutraceutical, herbal

## Abstract

Natural products with curative properties are gaining immense popularity in scientific and food research, possessing no side effects in contrast to other drugs. Guduchi, or *Tinospora cordifolia*, belongs to the menispermaceae family of universal drugs used to treat various diseases in traditional Indian literature. It has received attention in recent decades because of its utilization in folklore medicine for treating several disorders. Lately, the findings of active phytoconstituents present in herbal plants and their pharmacological function in disease treatment and control have stimulated interest in plants around the world. Guduchi is ethnobotanically used for jaundice, diabetes, urinary problems, stomachaches, prolonged diarrhea, skin ailments, and dysentery. The treatment with Guduchi extracts was accredited to phytochemical constituents, which include glycosides, alkaloids, steroids, and diterpenoid lactones. This review places emphasis on providing in-depth information on the budding applications of herbal medicine in the advancement of functional foods and nutraceuticals to natural product researchers.

## 1. Introduction

In order to open the door to an upcoming generation of therapeutic agents, there is a need to deduce the mechanisms of action that do not aim themselves as a substitute for medicines but are rather helpful in: (i) preventing a bunch of circumstances that might occur jointly (metabolic syndrome), like type 2 diabetes, stroke, and heart disease; (ii) balancing a pharmacological treatment for those individuals who fail conventional pharmacological therapy [1].

*Tinospora cordifolia* Miers (Willd.), acknowledged by the family of Menispermaceae, has been known by various names in India, such as Guduchi, Giloy, or Amrita, and in English, it is said to be a heart-leaved moonseed plant. It is very well renowned in Ayurveda and traditional medicine for its admirable therapeutic effectiveness [2] and is the richest source of many nutrients and phytoconstituents. It is comparable to ‘‘Nectar of Immortality’’ owing to its ability to confer juvenility, liveliness, and long life [3].

Across the world, 34 genus *Tinospora* species (Menispermaceae) are found all over Madagascar, Australia, the Pacific Islands [4], and tropical Africa [5]. *T. cordifolia*, a climbing deciduous shrub, is the only species indigenous to India and has been known for its varied therapeutic activities since ancient times. It is a shrub of high-altitude bearing flowers of green to yellow [6]. In the Indian Ayurveda system of medicine, *T. cordifolia* is usually recognized as an incredible herb possessing enormous pharmacological actions, such as antispasmodic, antiallergic, immunostimulatory [7], antidiabetic, anticancer, antineoplastic, anti-oxidative [8], antipyretic, antimalarial, antiosteoporosis [9], and anti-inflammatory [10]. In India, approximately 1000 tonnes are consumed annually [11]. All above said activities of this marvellous plant are credited for the existence of numerous categories of bioactive molecules, namely glycosides, diterpenoid lactones and alkaloids [2], sesquiterpenoid, steroids, and phenolics [12,13,14]. The plant is also rich in carbohydrate, fibers, and proteins. In small concentrations, vitamin C, calcium, and iron are also present [15].

## 2. Phytochemistry of *T. cordifolia*

The rich source present in *T. cordifolia* includes diterpenoid lactones, steroids, alkaloids, and glycosides [4,12,13,14]. Each part of the plant, i.e., leaves, stems, and roots possess a variety of secondary metabolites (Figure 1) with fascinating therapeutic applications (Table 1). Stems are significantly more significant from a therapeutic standpoint than the other sections of *T. cordifolia* [2,4,12,14,15,16,17,18]. Different phytochemicals, connected to its medicinal properties, are present in guduchi whole extract, including terpenes, alkaloids [3,15,19,20,21], steroids, glycosides, and flavonoids [22,23,24]. Due to its chemical composition, *T. cordifolia* has an unpleasant flavour that is bitter, pungent, and astringent; those chemicals are bitter diterpene, sesquiterpenoid, gilonin, glycoside, diterpenoid furanolactone, tinosporidine, palmarin and chasmanthin, columbin. As reported by Bernard et al. [25], the bitter flavour led folk medicine practitioners to increase metabolic activity, which cleanses cells, by eliminating the harmful metabolic byproducts.

The nutritional value of *T. cordifolia* was found to be 292.54 calories/100 gm. Kavya et al. 2015 [47] also reported that rich protein and dietary fiber contents found in gudichi have significant levels of major and minor elements, viz., Zn, Mn, Cl, K, Ca, Ti, Cr, Fe, Co, Ni, Cu, Br, and Sr, that act as micronutrients for health restorative purposes, and also play an important role in boosting the desired enzymatic activities.

### 2.1. Biological Activity

The bitter palate of *T. cordifolia* contributes to enhanced metabolic activity and destroys the harmful metabolic byproducts needed to purify tissues [4]. *T. cordifolia* extract acts as antistress and antidepressant agents; it increases anti-depressing monoamine levels, blocks monoamine oxidase, and inhibits the processes of gluconeogenesis and glycogenolysis [25,39,48,49]; it also induces hypoglycaemic effects [50], exhibited antidiabetic activity [51,52], has an immunomodulatory function, strengthens the immune system, and increases body resistance to illness [53,54,55]; it has been shown that HeLa S3 cells exhibit a cytotoxic effect against different cancer cell lines [56], and it possesses an antineoplastic effect. The extract also possesses antimicrobial activity against different microbial species [57] and has good hepatoprotective agents [4]. The plant parts, i.e., leaves and root extractives, have sturdy potential to scavenge free radicals [55,58,59,60] and aflatoxisis-related free radicals and nephrotoxicity brought on by nephrotoxins [61]. Stem extractive has been used to lessen the menace of diminished rat male fertility after oral consumption [39]. It is also found to be more effective in acute and sub-acute anti-inflammatory models than acetylsalicylic acid [62]. The *T. cordifolia* stem contains nutritional starch that is also used to treat a variety of illnesses (Table 2), regulate the level of blood sugar, stimulate the secretion of insulin, and help in the effects of insulin release and mimicking [63].

### 2.2. Pharmacological Properties

*T. cordifolia* is an herbaceous plant with multiple ensuring properties as a promising plant growth promoter, with impending benefits in poultry as food supplementation, as it has shown zero effect on DNA integrity, blood lymphocytes, and bone marrow [74]. Since the medieval times of Ayurvedic culture, *T. cordifolia* has been used to address metabolic issues and diabetes. The plant contains many chemical constituents like glycosides, alkaloids, phenolics, steroids, aliphatic compounds, diterpenoid lactones, and polysaccharides [67].

The plant decoctions and extracts possess various pharmacological functions, namely hepatoprotective [47], immunostimulatory [75], antipyretic [76], antidiabetic, and antiulcerative activity [25]. *T. cordifolia* in different species has helped pancreatic β-cells secrete insulin, which has led to its contribution to modulating carbohydrate and lipid-based mechanisms, resulting in massive quantities of fructose. The plant in the body has shown its positive effects in lipid metabolism, indicative of lipoprotein with densities of high, low, and very low, as well as refining hepatic acts [77], and SOD (superoxide dismutase) and glutathione peroxidase, as well as oxidative stress indicators and growing antioxidant enzymes in several models of animals, as shown in Table 3. Studies on *T. cordifolia* also demonstrated its antidiabetic benefits in rats, which were supported by a decrease in blood sugar levels. The methanolic extract has been shown to speed up the healing of wounds in albino mice [78]. Additionally, *T. cordifolia* was said to have analgesic properties, which were supported by a trial using albino rats [79]. As per various studies, the analgesic property involves both central (opioid receptors) and peripheral (prostaglandin synthesis inhibition) mechanisms [78,79,80]. The *T. cordifolia* possess various other pharmacological actions, as shown in Table 3.

### 2.3. Nutraceuticals

Stephen Defelice combined the words nutrition and pharmaceutical to create the phrase nutraceutical in 1989. He defines a nutraceutical as “any constituent that is a food or a constituent of food that affords therapeutic or health benefits, including disease prevention and treatment”. These goods can include everything from sequestered nutrients, nutritional additives, and a particular regime to exclusive foods made through genetic engineering and herbal remedies. The survey conducted in France, the United Kingdom, and Germany indicated that clients which practice nutrition assess higher or inherited aspects of achieving well-being [116,117]. The term “nutraceutical” was widely used in the United States, but there was no regulatory meaning. A functional food is defined as “a food that has a component incorporated into it to give it a specific medical or physiological benefit, other than purely nutritional benefit” by the Ministry of Agriculture, Fisheries and Food in the United Kingdom. The first stage in valuing nutraceuticals should involve a significant disparity between them and dietary supplements, as well as the recording of a suitable epidemiological goal.

Clinical trials on animals or in vitro are required in the pharmaceutical industry to confirm a compound’s effects. Contrarily, in nutrition, there has never been a mechanism for confirming the role of foods in the prevention or treatment of disease. However, since people’s awareness of health-related issues and the role that food may play in promoting good health and preventing disease has grown in recent years, the composition of food has been examined and verified scientifically. A wide range of therapeutic areas, including the treatment of cancer, depression, diabetes, cholesterol, blood pressure, and painkillers, can benefit from the use of nutraceuticals, including the treatment of cough and cold, anti-arthritis, digestion, sleeping difficulties, and pain relief. To find out how diverse nutraceuticals can be shown to be significant in the pharmaceutical industry, the research and development sectors for nutraceuticals are operating at their highest levels of productivity. Standardization of ingredients, cautious protocol creation, and implementation of clinical studies are all necessary due to scientific needs for nutraceuticals, which will have an impact on the industry and the health of consumers.

As was originally reported, food supplements have already been defined. Concerning nutraceuticals, the following definition was proposed: (i) a nutraceutical for food with a vegetal basis is a phyto-complex; and (ii) a nutraceutical is the collection of secondary metabolites for sustenance with animal origin. Both are administered in their appropriate pharmaceutical form and are concentrated. They may have positive health benefits, such as the ability to treat or prevent disease. The inclusion of nutraceuticals into regular diets may help prevent the onset of pathological conditions by possibly delaying or preventing the need for the use of pharmaceuticals in subjects who are qualified for an alternative, nonpharmacological strategy to a health condition.

There is a sizable knowledge gap about the advantageous benefits of *T. cordifolia* in the poultry sector, except for a few clear results concerning *T. cordifolia’s* good effects against *E. coli* in vitro. Therefore, with the help of the present review, the researcher’s investigation of drugs as natural growth promoters in poultry farming is advancing the understanding of pharmacology and strategizing into action. It is known for immune modulation, neuroendocrine, anti-inflammatory, antistress, antacid, antipyretic, and antioxidant effects [118]. Additionally, it elevated leucocytes counts, eliminated neutropenia, provided protection against infected rats and mice, cleansed the microcirculatory system and other physiological channels, and protected against external infection by removing toxins both exogenously and endogenously in a unique way [114,115,116,117]. The bitter taste, astringent property, thermogenic, digestive, anti-inflammatory, antipyretic, diuretic, uteric tonic, cardiotonic, and laxative are used in the treatment of jaundice, dyspepsia, cardiopathy, constipation, leucoderma, jaundice, renal and vesicles calculi, reduced triglycerides, and serum cholesterol in animals and in body weight. The probable mechanism was developed after going through various activities of individual phytoconstituents in the pharmacological studies, as shown in Figure 2 and Table 4.

Various constituents of *T. cordifolia* have been studied for immune-stimulant and immunomodulatory activity. The following mechanism has been deduced by various findings laid down by researchers: *T. cordifolia* resulted in increased production of hematopoietic factor GM-CSF (granulocyte-macrophages colony stimulating factor) by stimulating macrophages, which induced the enhanced production of leucocytes and suppressed chemotherapeutic neutropenia. Similarly, a novel glycoside (1,4-α-D glucan) activated the immune system through macrophage activation via the TLR signaling pathway and increased cytokine production. Further, it stimulated humoral immunity by increasing antibody-producing cells and circulating antibody titers. The active constituent also stimulated phagocyte activity. The constituents isolated from the plant also showed cutaneous reactions contributing to lymphokine liberation and T-cell-sensitized chemotactic factor. It also possessed immune-potentiating activity, as evidenced by increased IgG antibody production. The polysaccharide (G1-4A) has shown action against the inhibition of *Mycobacterium tuberculosis* survival extracted from plants. The polysaccharide component present in plants modulates the host immune response through the TLR4-dependent pathway, leading to increased B cell production and controlling and balancing inflammation in response to pathogens. The pre-administration of polysaccharide prohibited lipopolysaccharide (LPS)-induced mortality, which was co-related with a reduced TNF-α response, increasing circulated levels of TNF receptors, and declining nitric oxide release by splenic adherent cells. This has also lessened pulmonary bacillary burden, which is coordinated with reduced Th2 cytokines and elevated Th1 cytokines. It has also positively inhibited surface expression of MHCII (major histocompatibility complex-II) and CD-86, eliciting a TLR4-MyD88-dependent Th1 cytokine response characterized by upregulation of TNF-α and IL-1β. The pharmacologic inhibitors showed the physiological key role of cell signaling pathways, including p38, ERK, and JNK MAPKs, in macrophage activation by polysaccharides.

The populace has access to traditional drugs, but many use nutraceuticals, including herbal remedies and other nutritional supplements, whereas older adults exhibit high polypharmacy. Thus, the interaction of poly-component nutraceuticals and drugs is a subject of utmost importance. To evaluate the nutraceutical-drug interactions experimentally, factors like history of safe usage, literature evidence from animal toxicity and human clinical research, and nutraceutical ingredient characterization might provide guidance. Different regulatory agencies have approved and registered chemical compounds of *T. cordifolia*, namely berberine and tinocordifolin, for certain therapeutic purposes.

### 2.4. An Insight for Application in Food Products

Due to the gigantic medicinal products discussed above, *T. cordifolia* has an exclusive place in human food science. The literature suggested that the drug is chiefly utilized in the raw form as syrup and as nutritional additives in various phases of disease infections like in the case of dengue fever [124]. Its application as a preventive ingredient in the food system is undeniable, and its prophylactic constituents are also equally noteworthy. Due to its bitter taste, it lacks commercialization in food-incorporated preparation in the market, which prohibits people from consuming it in their daily diet, and has led to certain challenges [125]. While going through many databases, very limited scientific information was available to collect on its application in food due to the high sympathy to its characteristic taste and tainted aroma like in milk and meat products. Thus, it is advisable to investigate the processing techniques and experiments for its active employment in food products that have intact medicinal possibilities that are well-studied in food science.

#### 2.4.1. Various Ways of Using *T. Cordifolia* in Foods

Like other medicinal plants, *T. cordifolia* was also investigated for its therapeutic potential by consuming either the parts like leaves, roots, stem, or the whole plant in the preparation of functional foods. All plant parts are utilized for the formation of medicinal powder, juices, and extracts. Although all individual parts contributed to the notable and beneficial well-known therapeutic characteristics searched and recapped, stem is used effectively and establishes higher pharmaceutical values [126].

Various solvents have been used for extraction of *T. cordifolia* stem for finding bioactive polyphenol components and further investigating its antioxidant potential. Methanol and ethanol extract showed the presence of vanillin, arbutin, trans-ferulic acid, caffeic acid, trans cinnamic acid, catechin-hydrate, epicatechin, and ellagic acid, whereas polyphenol, flavonoids, and tannin were present in methanolic extract. Due to the abundance of pharmacologically active components, *T. cordifolia* might be targeted as a potential functional component in developing functional foods. The leaf powder in some research has shown higher solubility and antioxidant potential than the stem powder [127]. The powder was prepared by spray drying, which led to mass transfer, and minimal temperature resulted in sting shell and accumulation in stem powder, which initiated higher moisture content, total phenolic, and antioxidant content, and the powder became hygroscopic as compared to the leaf powder. Hence, in our understanding, more research and experimentation should be performed to identify and quantify the bioactive compounds obtained by using spray drying, formulating functional food.

The addition of *T. cordifolia* in food products unquestionably will significantly contribute to sustaining human health along with providing customers with varied arrays of eatables, thus heightening the nutritional values in food. Thus, prior to the addition in any form (namely juice, powder, and extract), certain challenges need to be mentioned meritoriously in a way that the presence does not vary the sensory quality, particularly in food products, in terms of its flavour and appearance. The incompatibilities like bitterness, renneting and fermentation characteristics, and lowly heat processing ability need to be studied extensively and scrutinized for its efficacious use in food commodities [128]. It is very well known that most active pharmaceuticals in any herbal drug contribute to the strong bitter aftertaste. Thus, the requirement for a thorough examination for finding alternative methods and processes for removing the bitter properties of juice, powder, and extract formation from the plant.

#### 2.4.2. Debitter Methodology in Food Products

Various methodologies and procedures were tried and tested for removing bitter and stabilizing acerbic juices like *Embilica officianalis*, *Momordica charantia*, *Azadirachta indica* [129]. These approaches can also be utilized in the elimination of bitterness in *T. cordifolia* formulations, or else the palatability of the drug decreases in incorporated food products. However, recent reports have shown none of the methods used by researchers have been successful in debittering *T. cordifolia* juice [13]. To debitter *T. cordifolia* powder, many membrane processing techniques like ultrafiltration and microfiltration were explored, but they failed to obtain any significant results. The physico-chemical composition did not change with these techniques and the total phenol content was also retained simultaneously by the microfiltration method. Rawson and his co-worker [130] investigated the possible application of thermosonication on bioactive compounds of fresh watermelon juice such as ascorbic acid, total phenol, and lycopene as a function of treatment temperature (°C), amplitude level (μm), and processing duration (min). The research reveals the importance of watermelon juice quality characteristics, which are greatly influenced by thermosonication, and further, the process variable parameters were optimized by the response surface methodology (RSM) to produce juices with high bioactive ingredient retention levels. Processing stability retained bioactive components by releasing bounded phenolics after its exposure to sterilization temperature. After so many hits and misses in debittering techniques hypothesized by authors (Figure 3), a big challenge persists for the food processing industries to design more effective techniques to eliminate bitterness in *T. cordifolia* products below its permissible limits for increased acceptability among consumers.

#### 2.4.3. Functional Ingredient in Food Products

With advancements in science and technology, the increased health awareness among consumers has put immense pressure on the research community to articulate and design diverse food products with better nutritional and functional properties [81]. Nonetheless, the use of herbal drugs in foodstuffs upgraded the nutritional attributes and increased the bioavailability of food products.

Previous analyses mainly focused on confectionary and bakery products. In current trends, fewer researchers have also focused on using commercially available *T. cordifolia* formulation in pills and as extractives to investigate its impending benefits in focusing on the fat oxidation issue in meat products. The study was performed on *T. cordifolia* stem powder to examine its therapeutic potential in developing nutritive cookies, where the concentration was varied from 0 to 12% [126]. The *T. cordifolia* stem powder was incorporated in cookies in the range of 0–8% due to the presence of bitterness beyond 8.0%. Thus, 8.0% limits the desirable addition level in bakery products. The anti-oxidant potential (total phenolic and flavonoid amount, DPPH, nitric oxide scavenging and Ferric Reducing Power Assay) was augmented in cookies fortified with *T. cordifolia*. The baking process in nutritional cookies brings insignificant modification to the functional properties, although further investigation should be carried out to incorporate high levels of *T. cordifolia* powder to explore its full efficacy and to identify changes in active phytoconstituents after baking. Tyagi and his co-worker [126] investigated the presence of the bioactive components present in *T. cordifolia*-incorporated cookies using advanced and non-conventional methods. Cookies were prepared with defatted *T. cordifolia* at different time intervals. No significant changes were observed in extraction time on the concentration of active constituents by adsorption chromatographic fingerprinting at low and high wavelengths, confirming the existence of some constituents with similar Rf readings in all the prepared samples but in reduced concentrations [131]. Thermogravimetric analysis (TGA) was also performed to check the processing consistency of *T. cordifolia* components while baking. The TGA results revealed that there was no change or loss in the composition of the cookies at 220 °C, the final melting temperatures of cookies. In comparison to the control, Sood and Shilpa’s [125] showed FT-IR analyses of biscuits fortified with *T. cordifolia* revealed the presence of amine and amide groups. These groups might denote the existence of berberine as a segment of *T. cordifolia*’s alkaloid ring. Thus, these investigations on bakery goods fortified with *T. cordifolia* provide a scope to researchers to further discover the principal compounds by LC-MS and NMR and its utilization in liquid preparations like squash [125].

The addition of *T. cordifolia* poses stability and low bioavailability issues in any food products. To resolve this issue, an appropriate *T. cordifolia* delivery system was founded by researchers employing the collaborative potential of bioactive phytoconstituents (berberine, palmatin, flavonoids, cardiac glycosides, etc.) [132]. Electron spray and encapsulation techniques were used for providing controlled release of the antidiabetic effect of extracted bioactive compounds from *T. cordifolia* in a whey protein isolate’s shell matrix. It was used to fabricate the nanosphere and increase antidiabetic potential by 28.12% in encapsulated form after a day. Further, studies were conducted on investigating the effect of controlled release of phytopharmaceuticals in various rodents for determining their usage as functional components or nutraceuticals.

The goat milk fortified with giloy as a beverage was processed, considered as ‘green milk’ and targeted as an ideal vehicle for bioactive molecules for a drug delivery system. The study was conducted by Sharma and co-workers to develop a functional drink, fortifying goat milk with giloy juice [128]. An upsurge in total solids content, particle size, and antioxidant activity were observed in the beverage due to the incorporation of *T. cordifolia* juice with massive therapeutic functions. With the increase in shear rate, the viscosity of the beverage reduced, leading to shear thinning behaviour. Scanning electron microscopy was done on milk protein fortified with *T. cordifolia* to study its polyphenol interactions. It was found that the 5 days in refrigeration led to the shelf life of pasteurized giloy-goat milk having better sensory quality and increased retention of pharmacological potential.

Meat and meat products are other categories of foods where *T. cordifolia* established itself as a potential candidate. It is well known that the depreciation change occurred in the meat products due to oxidation of lipids, which is easily controlled by the incorporation of polyphenols and flavonols from *T. cordifolia* extracts as a potential conserving agent in meat products. Kalem et al. [133] reported the addition of readily available commercial *T. cordifolia* pills in meat suspension at different concentrations (0.25%, 0.5%, and 0.75%). Further, to evaluate the efficacy and storage stability of pills, chevon sausages were prepared using *T. cordifolia*-treated emulsion. With the increased amounts of flavonoids and polyphenols, antioxidant and antimicrobial properties were higher, whereas thiobarbituric acid reactive substrate and free fatty acid content were less than controlled samples. To enhance the stability of treated chevon sausages during storage, active components of *T. cordifolia* helped in the decomposition of peroxides along with free radical adsorption and neutralization with the quenching of singlet oxygen. Antibacterial and antifungal studies were performed by reduced psychrophilic count, plate count, mould, and yeast count. Natural antioxidants present in *T. cordifolia* capsules delayed the protein degradation and lipid breakdown, hence improving suitability and refining the flavour in meat products. The use and function of *T. cordifolia* stem was tested with butylated hydroxytoluene (0.02%) as a synthetic antioxidant to demonstrate its potency in hen meat patties [13]. To monitor the oxidative stability, meat patties were incorporated with *T. cordifolia* at a concentration of 3.0% and 4.0% and stored at refrigeration temperature for testing. Increased total flavonoid content and high DPPH along with reduced microbial count and lipid oxidation parameters were found in *T. cordifolia* stem in presence of polyphenolic compounds. Several other studies were also conducted on the sensory evaluation scores, which showed good oxidative stability (at 3.0% level) and storage stability in hen meat products. *T. cordifolia’s* potency as a PPL (porcine pancreatic lipase) antagonist and antioxidant was examined in nuggets prepared by goat meat using an in vitro digestion model [134]. Different prepared alcoholic and aqueous extracts were tested for various activities, where the key substrates used were 2,4-dinitrophenyl butyrate (DNPB) and tiolein. In ethanolic extract, higher PPL inhibition action was observed in polar and semi-polar phenolic acid soluble with good cell wall degradation ability. Fatty acids like oleic acid, elaidic acid, lauric acid, linolenic acid, myristoleic acid, eicosapentanoic acid, and linoleic acid showed higher values, whereas palmitoleic, palmitic acid, stearic acid, and heptadecanoic levels reduced in meat products after digestion by in vitro analysis. The results were compared with positive control Orlistat, hypothesizing that *T. cordifolia* extract might act as a hopeful option for a manmade PPL inhibitor and as an antioxidant.

*T. cordifolia* rich in bioactive components has also been explored as edible packing material in food products. In the year 2018, a unique eco-friendly and bioactive edible film layer of calcium alginate was effectively processed to enhance and upgrade the storage, stability, and shelf life of meat products [133]. For the testing of *T. cordifolia*-fused edible film, the model chosen for the study was goat meat sausages. The drug capsules were administered at different concentrated levels (0.5% and 1.0%) in calcium alginate film based on maltodextrin enfolded all over the sausages by shedding off the coverings following refrigeration. The antimicrobial and antioxidant potential in examined sausages were improved in a concentration-dependent manner as free fatty acid and thiobarbituric acid substrates were lowered in edible film. The organoleptic and the sensory properties were enhanced in goat meat sausages by a spoilage hindering method. Thus, this review put forward enormous unexploited prospectives in discovering the various ways of using *T. cordifolia* in formulating functional food and pharmaceutical products, boosting the market sector profits, and will be beneficial for people of all domains.

## 3. Conclusions

Since ancient times, *T. cordifolia* has been known for its therapeutic properties and has crossed several eras from historic to the current generation and accomplished glory as a miraculous plant, having an enormous therapeutic function in the modern age, providing immense health benefits to consumers. The active phytoconstituents present in *T. cordifolia* belong to the class of alkaloids, glycosides, terpenoids, steroids, polysaccharides, etc., for its pharmacology. There is a wealth of evidence supporting the drug’s antiangiogenic, cardioprotective, immunomodulatory, antidiabetic, antiallergic, and neuroprotective effects, which have been generated by study through in vivo experiments and a few supportive clinical trials. Although the underlying mechanism convenes to TLR6 signalling, macrophage activation, increased antioxidative enzymes, and modulating pancreatic and cytokine production, etc., a clearcut mechanism is still ambiguous. Various methods were followed for the isolation of active phytoconstituents of *T. cordifolia* along with purification, which were accountable for treating different diseases. The research should focus primarily on the *T. cordifolia*-fused product because the raw form of the plant in various states as powder and extract has a very unpleasant and bitter taste. Firstly, retaining active phytoconstituents responsible for activity in food products is crucial and requires in-depth investigation. Secondly, the stability of drugs incorporated in food products is further studied by using various emulsion forms. Investigators should also explore various flavour elements or natural debittering to enhance the acceptability in the food industry. It is crucial to authenticate intuitions of the therapeutic efficiency of *T. cordifolia*-induced food products with clinical and preclinical studies. All these parameters will help in expanding the horizons for uplifting the market revenues, and aid in accomplishing the ambitions of customer agreement along with product diversification. With the introduction of functional food and nutraceuticals in the market as prophylactic agents in the food sector, adding drug formulations in juice, powder, extract, and capsule forms into different food sectors has shown to reduce the risk of oxidation of lipids and fats and improve the sensory quality and utility of the consumable products. The fortification of *T. cordifolia* in eatables is growing and has vast business potential, with abundant prospects to explore various stabilizing techniques for *T. cordifolia* formulations at amplified levels in pharmaceutical and food products without negotiating the sensory attributes.

## Figures and Tables

**Figure 1 molecules-28-07073-f001:**
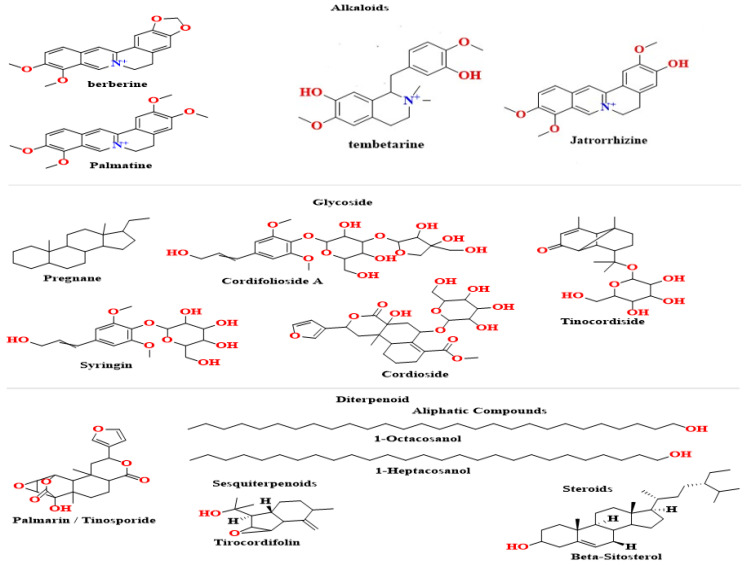
Chemical structures of active compounds present in *T. cordifolia* stem.

**Figure 2 molecules-28-07073-f002:**
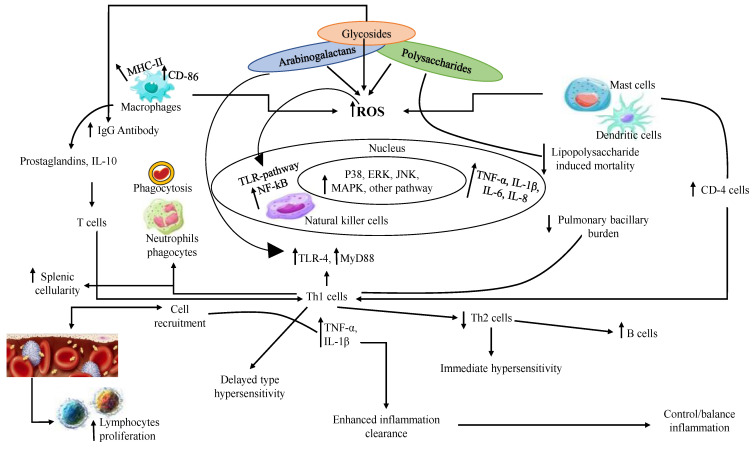
Probable mechanism of action underlying immunomodulation.

**Figure 3 molecules-28-07073-f003:**
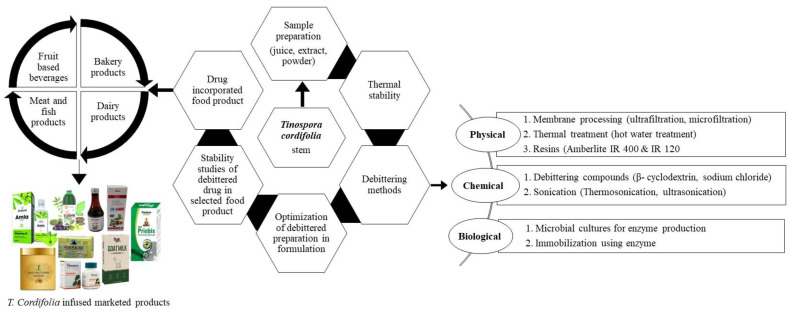
A pictorial view of the steps involved in addition of *T. cordifolia* in food products.

**Table 1 molecules-28-07073-t001:** Major chemical constituent class and their active phytoconstituent with pharmacology.

Chemical Class	Active Constituents	Phytoactive (Class)	Activity	References
Glycosides (Phenylpropanoid)	Tinocordioside, glucoside 18-norclerodane, Pregnane glycoside, diterpene Furanoid glucoside, Cordioside, Tinocordifolioside, Syringin, Cordifolioside A, B, C, D and E, Palmatosides, Syringinapiosyl glycoside, 2-Methyl-1,2-pyrrolidine, Cordifolioside, N-Formylannonai	Cordioside (Clerodane furano diterpene) Syringin, Cordifolioside, (phenylpropanoid glycoside)	Enhance phagocytosis	[22,26,27,28,29]
Alkaloids	Berberine, magnoflorine, aporphine alkaloids, jatrorrhizine, tembetarine, tinosporin, isocolumbin, tetrahydropalmatine, choline, palmatine	N-Formylannonain, Magnoflorine (Benzylisoquinoline alkaloid)	Increment in ROS, phagocytosis	[30,31]
Diterpenoid lactone	Furanolactone, Tinosporon, Diterpenoids, Tinosporides, Tinosporon, Jateorine, columbin, Clerodane derivatives	-	-	[32,33,34,35,36,37]
Steroids	β-sitosterol, 20 β-hydroxyecdysone, Makisterone A, Giloinsterol	-	-	[38]
Sesquiterpenoids	Tinocordifolin, einocordifolin	Tinocordiside, 11-hydroxymuskatone (Cadinane sesquiterpene)	Increased phagocytosis, increase ROS	[30,39,40]
Aliphatic compounds	Octacosanol Heptacosanol, Nonacosan-15-one dichloromethane	-	-	[40,41,42]
Lignans	3,(a,4-dihydroxyl-3-methoxy-benzyl)-4-(4-hydroxy-3 methoxy-benzyl)-Tetrahydrofuran	-	-	[32]
Other compounds	Tinosporidine, Arabinogalactan, Giloin, Nonscosan-15-one, Jatrorrhizine, Cordifol, Cordifelone, Giloinin, diacetate N-transferuloyltyramine, Tinosporic acid	Arabinogalactan polysac (G1-4A)	Decrease mortality, increase Th1, decrease Th2, increase macrophage activation, increase mitogenesis	[23,41,43,44]
Polysaccharide	Alpha-D-glucan	RR1	Increase phagocytosis; increase Th1 cytokines	[45]
Protein	Guduchi immunomodulatory protein (ImP)	-	Increase phagocytosis; increase mitogenesis, increase bactericidal	[46]

**Table 2 molecules-28-07073-t002:** Summary of critical studies accompanied by different pharmacological studies using different *T. cordifolia* samples.

Extracting Solvent	Bioassay Method	Observation and the Outcome of the Study	References
Aqueous extract (Aqs Ext)	SOD, GPx activity, β-Glucuronidase, catalase, LPO	By supplementing catalases, SOD, GPx, β-glucuronidase, LPO, and oxidative stress are significantly reduced. By boosting ROS enzymes, denervation markedly increased oxidative stress. Catalase and GPx activities in denervated mice reduced. An excessive ROS load in the muscles. Denervated mice with increased SOD activity and antioxidant potential	[12]
Alcoholic (Alc) and Aqs Ext	Catalase	In diabetic rats induced with alloxan, catalase activity and erythrocyte membrane lipid peroxide were increased. Decreased glutathione peroxidase and superoxide dismutase.	[64]
95% Ethanolic (Eth) Ext	LPO, SOD	Significant increase in enzymatic and non-enzymatic antioxidants of hepatocellular carcinoma in animals Decreased LPO limits	[65]
Aqs and Alc Ext	DPPH	Methanol extract exhibited substantial antioxidant activity in vitro. Possess high antioxidant potential	[66]
Eth and meth Extfrom stem bark	DPPH	Reduced free radical scavenging potential in methanolic bark tract as compared with 71.49% ethanolic extract.	[42]
Meth Ext	DPPH	Compared with standard ascorbic acid showing IC_50_ (5 μg/mL) having potent antioxidant activity and inhibitory concentration	[67]
Aqs Ext	Myeloperoxidase (MPO), creatine kinase (CK) assay	Denervation considerably upsurge inflammation by boosting inflammatory enzymes. Denervation induce inflammation for anti-inflammatory enzyme by obstructing protective mechanism	[16]
Aqs Ext	MPTP-intoxicated Parkinsonian mouse model	Reversal of major behavioural and biochemical deviations in mice model Inhibition of activated MPTP-intoxicated NF-κB with associated pro-inflammatory cytokine TNF-α. Dopaminergic neurons shown protection by suppressing neuroinflammation in Parkinsonian mouse model	[68]
Composite Aqs Ext of *T. cordifolia* and *Z. officinale*	MTT assay	Effect of composite extract in MCF-7 cells is more effective having IC_50_ 2 μg/mL than the individual drug, i.e., T. cordifolia and Z. officeinale at IC50 509 μg/mL and 1.0 mg/mL, respectively Network analysis and in vitro validation showed synergistic effect in breast cancer	[69]
Meth Ext	Vital dye Presto Blue-based assay	33 genes involved in Berberine therapy inhibited the expression involved in the progress of differentiated cells during diverse differentiation and epithelial-mesenchymal transition in dose- and time-related processes. Berberine supported two medication forms and varied complete dosages. Hence, berberine present in the drug possesses antiproliferative activity.	[18]
Chloroform andhexane extracts	MTT assay	Decreased the proliferation rate significantly inducing cell differentiation. Delay in proliferation rates, resulting in suppression of migration of human glioblastomas and neuroblastomas.	[20]
Meth Ext	MTT assay	Demonstrated antiproliferative action in MDA-MB-231, and MCF-7 (breast cancer cells) further stopped the growth. ROS inhibited the growth of breast cancer cells and promoted apoptosis. ROS plays an important role in the apoptosis of cancer cells by extract Quercetin and Rutin are being studied for anticancer potential against these cancer cells	[70]
Aqs-Eth Ext (50%)	MTT assay	Cytotoxic effects shown on IMR-32 cells. Anti-cancerous, differentiation, and metastasis inhibitory actions of combination active phytoconstituents in neuroblastoma cells are demonstrated. The higher the concentration of treatment cells, the more they can be elevated and the fate of terminally differentiated cells can be traced.	[71]
Methanol and acetone (70:30) extract	DMBA-induced skincancer model	Consumption of Palmatin orally in the daily routine might inhibit environmental carcinogenic substances causing skin carcinogenesis, hence offering protection against skin cancer.	[72]
Hydro-ethanolicextract (50%)	Proliferation assays	Protection against cell proliferation and growth.	[71]
Methanolic extract of fresh stem (50%)	Melanoma assay	Improvement in the lifespan of C57 Bl mice. Tumour size substantially minimized relative to control with 50% hydroalcoholic extract at a dosage of 750 mg/kg body wt. for 30 days.	[73]

**Table 3 molecules-28-07073-t003:** Detailed overview of the studies presenting the efficacy of various preparations of *T. cordifolia* against certain diseases.

*T. cordifolia* Preparation	Animal Model Used	Dose Administered	Study Objective	Experimental Results	References
**Immunomodulatory Activity**					
Aqs Ext	Swiss albino mice	15 days-100 mg/kg/day	Colony-stimulating activity	Predominant neutrophilia with induced leucocytosis	[60]
Arabinogalactan from Aqs Ext	Spleen cells of murine	-	Mitogenic activity	Increased mitogenic activity	[23]
Methanolic extract	Balb/c mice	5 days-200 mg/kg/day	Phagocytic activity antibody production	Enhanced phagocytic activity, increased WBC, increased immune response, increased stem cells maturation	[80]
Ethanolic and Petroleum ether extract	*Oreochromis mossambicus* (vaccinated with heat killed *A. hydrophila*, 109 cells/fish)	0.8, 8 or 80 mg/kg	Neutrophil activity antibody response	Prolonged the peak primary antibody titres (1–3 weeks); enhanced 2° antibody responses (8 mg/kg) and neutrophil activity	[81]
08 compounds (*N-formylannonain*, *magnoflorine*, *jatrorrhizine*, *palmatine*, *11-hydroxymustakone*, *cordifolioside A*, *tinocordiside* and *yangambin*)	Primary cells of murine	Isolated molecules of 10, 25, 50 and 100 μM	Splenocyte assay	Immunomodulatory activity possesses by *N-formylannonain* and 11-*hydroxymustakone*	[30]
Aqs Ext	Cholestasis-induced rats	7 days-100 mg/kg	Macrophage activity, cellular immune functions, and polymorphonuclear cells	Improved cellular immune function, 16% reduction in mortality rate by *E. coli* infection, enhanced phagocytic cell functions	[82]
*T. cordifolia*, *T. chebula*, *B. diffusa*, *B. aristata*, and *Z. officinale* Eth Ext	Golden hamsters (inoculated with E.Histolytica trophozoites)	4 days-400, 600 and 800 mg/kg	T and B cell counts and hemagglutination titre	No effect on T-cell counts, enhanced cell mediated immunity, increased humoral immunity	[83]
Polysaccharide isolation (α-D-glucan)	HEK 293 cell lines, Mouse macrophages (RAW 264.7)	0, 100, 500 and 1000 μg/mL	NF-κB enhance activated B cells, TNF-α synthesis; opsonic binding and phagocytosis	TNF-α synthesis inhibition of macrophages cell line, phagocyte inhibition, activated NF-κB in dose and time-dependent manner	[45]
G1-4A polysaccharide	primary murine macrophages and RAW 264.7 cells from BALB/c mice	Constant 8 h-1.0 mg/mL	NO and cytokines production, *M. tuberculosis* intracellular survival, and phagocyte assay	Increased NO level, activated macrophage, IL-1, IL-2, and TNF-levels upregulation, increased MHC-II and CD86	[84]
Aqs Ext	Immunized Albino rabbits with Typhoid ‘H’ antigen	20 days-10 mg/100 g	Antibody titre	Reduced antibody formation and immunosuppressive action	[85]
80% Eth Ext	Induced Ochratoxin A Albino mice of Hindustan Antibiotics strain	17 weeks-100 mg/kg	Cytokine production, Macrophage chemotaxis	IL-1α and TNF-α production, inhibition of suppressed chemotactic activity	[86]
Non-polar, alkaloid-free extract and polar fraction	Ascitic tumour-induced BALB/c mice	15 days-100 mg/kg	Serological and hematological parameters, antibody titres	Ineffective haematological parameters and myelo-protection, increase antibody titres	[87]
**Antidiabetic Activity**					
13 active compounds	computational studies	–	Glycogen phosphorylase activity	Glycogen phosphorylase activity is decreased by magnoflorin, cordiofolioside A, and syringin.	[88]
Aqs Ext	Alloxan rats	21–120 days-dose 400 mg/kg	Antihyperglycemic	Decreased amounts of the substrate and the enzymes hexokinase, phosphofructokinase, and glucokinase	[47]
Aqs and Alc Ext	Swiss albino mice (injected with Ehrlich ascites tumour cells)	10 days-1–100 μg	Glucose uptake under tumour conditions	Ethanolic extract (100 μg) and methanolic extract (40 μg) showed good glucose uptake	[89,90]
Alc and Aqs Ext Stem	Streptozotocin-induced diabetic albino rats	30 days-200 and 400 mg/kg, respectively	Enzymes involvement in glucose metabolism, Serum insulin level	When compared to insulin, it is 40–80% more effective; it also increases hepatic glycogen synthase and decreases glycogen phosphorylase activity.	[91]
Extraction using different solvents	Albino rats induced by alloxan	1.0 day-0.3 mg/g	α-glucosidase inhibition	Reduced post-meal spike in blood sugar, Salivary amylase (75%), -glucosidase (100%), and pancreatic (83%) activity are all non-competitively inhibited.	[92]
Alkaloid-rich fraction	Tolbutamide-induced diabetic wistar rats	Magnoflorine, palmatine, and jatrorrhizine (10, 20, and 40 mg/kg each), Isoquinoline (50, 100, and 200 mg/kg)	Antihyperglycemic	Blood glucose levels are kept from increasing with a decreased fasting serum glucose and a glucose supplement of 2.0 g/kg.	[63]
Aqs Ext	Albino Wistar rats given with high fructose diet	60 days-400 mg/kg/day	Carbohydrate metabolism	Reduced the rise in triglycerides (54.12%), insulin (51.5%), glucose-insulin index (59.8%), and blood sugar levels (21.3%).	[93]
Alc and Aqs Ext	Streptozotocin diabetic rats	40 days-400 mg/kg	Antiglycemic effect	7.45% lessened in plasma glucose level, prevented polyuria	[94]
**Nephroprotective Activity**					
Meth and Aqs Ext	Cyclophosphamide-induced Swiss albino mice	5 days-200 mg/kg	Urine protein and urea nitrogen content, serum cytokine level, urinary glutathione content	Reduced TNF-α level, increased glutathione level in bladder and liver, decreased protein level in urine and serum, enhanced IL-2 and IFN-gamma levels	[61]
Alc and Aqs Ext	Streptozotocin-induced diabetic rats	40 days-400 mg/kg	Renal damage assay	Considerably controlled urinary albumin levels; no impact on renal hypertrophy	[47]
**Anti-Inflammatory Activity**					
Solution of powder in 2% gum acacia	1.0% suspension of carrageenan in normal saline; sub-plantar injected in albino rats	6 days-50 mg/kg/oral	Acute and chronic inflammation	67% inhibition in granulation tissue of paw in both acute and sub-acute inflammation	[95]
Aqs Ext	For arthritic syndrome, albino rats were given 1.0% carrageenan and 2.0% croton oil in ground nut oil, respectively, along with 0.1 mL of Freund’s adjuvant.	Acute inflammation-60 mg/100 g, chronic inflammation-20 mg/100 g, arthritic syndrome-10 mg/100 g	Mild analgesic effect, mean volume of edema, percent inhibition of edema	Dose-dependent effect, minimal impact on volume of edema, oral drug administration results in a 63.16% edema suppression compared to intraperitoneal injection’s 49.20%	[85]
**Antineoplastic Activity**					
Methanolic, aqueous, and methylene chloride stem extract	HeLa S3 cells	0, 5, 10, 25, 50, and 100 mg/mL	Micronucleus assay, cytotoxicity, Colonogenic assay	Concentration-dependent increased frequency of micronuclei, dose-dependent increase in cell killing by 2.8 (50 mg/mL) and 6.8 (100 mg/mL)-fold, reduction in survival fractions of cells	[96]
Berberine-rich extract	Human tongue squamous carcinoma SCC-4 cells injected in BALB/cnu/nu nude mice	28 days-10 mg/kg	Tumour size and volume	Significantly lower tumour volume and growth (52% tumour inhibition)	[97]
Eight isolated compounds from Eth Ext	HT-29 (human colorectal cancer), SiHa, CHOK-1 (hamster ovary), and KB (human oral squamous carcinoma)	Molecular isolates at 10, 25, 50, and 100 μM	Cytotoxicity (Sulforhodamine B assay)	All are effective against KB and CHOK-1 cells, while yangambin, palmatine, and tinocordiside are also effective against KB and HT-29 cells and KB and CHOK-1 cells, respectively.	[30]
Eth Ext	C6 glioma cells	250 μg/mL and 350 μg/mL	Cytotoxic and antiproliferative property	Enhanced production of mortalin, differentiation in C6 glioma cells, cell proliferation reduction in dose-dependent	[98]
Epoxy clerodane Diterpene	Diethyl nitrosamine induced liver tumour in Wistar albino rats	20 weeks-10 mg/kg	Biochemical parameters, liver morphology	Reduced liver weight, decrease in lactate dehydrogenase, catalase, glutamate transaminase and pyruvate transaminase	[99]
Meth Ext (70%)	Mice C57BL/6 (angiogenesis induced with metastatic B16F-10 melanoma cells)	5 days-20 mg/kg	synthesis of cytokines, antiangiogenic compounds, and growth factors	Metalloprotease-1 tissue inhibitor synthesis was increased, and tumour-directed capillary development was suppressed. Pro-inflammatory cytokines such as granulocyte-monocyte-colony-stimulating factor were decreased.	[100]
Polysaccharide fraction	Mice C57BL/6 induced with metastatic B16F-10 melanoma cells	0.5 mg/dose/animal	Enzyme level in serum, biochemical parameters, survival rate,	Decrease in lung collagen’s hydroxyproline, hexosamines, and uronic acids; inhibition of lung metastases (72%); reduction levels of sialic acid and gamma-glutamyl transpeptidase	[101]
**Antioxidant Activity**					
Stem powder	Streptozotocin-induced CFT-Wistar pregnant diabetic rats	1.0% and 2.0%	Embryo-lethality, oxidative stress markers in fetal brain and liver	53% and 48% increment in glutathione and total thiols level, 63% protection against embryo-lethality, 25% and 72% decline in malondialdehyde and reactive oxygen species, respectively	[102]
Stem Aqs Ext	High fructose diet given to Albino Wistar rats	400 mg/kg/day (for 60 days)	Oxidation stage	34% and 28% reduction in TBARS and protein carbonyl groups, respectively	[103]
**Antistress Activity**					
Aqs Ext	Sprague Dawley rats	15 days-100 mg/kg	Urine metabolites	Regulation of glomerular filtration rate, TCA cycle, Gut Microflora activity and catecholamine pathway	[104]
Diterpene isolated Epoxy clerodane	Wistar albino rats (indomethacin-induced gastric ulcer)	Single dose-12.50, 25, and 50 mg/kg	ulcer index, myeloperoxidase activity, and gastric mucosal lesions; anti-inflammatory and pro-angiogenic factors	Increased gastric myeloperoxidase activity, 3.01- and 2.26-fold increase in IL4 and IL-10, respectively, whereas 1.67-, 1.72-, and 1.71-fold reduction in TNF-α, IL-1, and IL-6, respectively, 1.5-fold pro-angiogenic growth factors increase, 54.42% apoptotic index, reduction in ulcer index and gastric mucosal lesions (91.80%)	[105]
**Hepatoprotective Activity**					
Stem decoction	Horse serum injection in rats	30 days-100 mg/kg	Kupffer cell activity performed by carbon clearance method	Prevention of suppression of Kupffer cells of liver in treated groups by lowest carbon half-life	[82]
Stem Aqs Ext	Wistar rats (pyrazinamide, isoniazid, and rifampicin induced hepatic damage)	90 days-100 mg/kg	Hepatic enzymes, liver morphology	Restriction in weight and liver volume, together with normal levels of bilirubin, alkaline phosphatase, aspartate transaminase, and alanine transaminase	[106]
Aqs Ext	Wistar strain of Albino rats (intoxicated with carbon tetrachloride)	100 mg/kg (for 15 days)	Enzyme level in serum	Reduction in serum glutamate oxaloacetate transaminase, alkaline phosphatase, and bilirubin level (2.2 mg/100 mL)	[54]
**Antiparasitic Activity**					
Eth Ext	*Pediculus humanus capitis*, *A. subpictus* and *C. quinquefasciatus*	100 mg/L	Larvicidal, pedulocidal	75–90% mortality against *A. subpictus*, *C. quinquefasciatus larvae*, and in *Pediculus humanus capitis* adults	[107]
Guduchi satwa	Paracetamol induced hepatotoxicity in Wistar albino rats	4 days-200 and 400 mg/kg	Blood biochemical markers	High levels of alanine aminotransferase, aspartate aminotransferase, alkaline phosphatase, and total bilirubin were reduced.	[108]
Whole plant extract	BALB/c mice (infected with 107 promastigotes of *L. donovani*)	15 days-100 mg/kg	Tests for liver and renal function, parasite load, immunoglobulins, cytokines, and delayed hypersensitivity	Significant increment in delayed hypersensitivity, reduction in hepatic parasitic load, increased IgG production with no effect in SGOT and SGPT levels	[82]
**Hypolipidemic Activity**					
Aqs Ext	Wistar rats (high fructose diet)	60 days-400 mg/kg/day	Lipid enzymes	Attained adipose tissue fatty acid synthetase, malic enzyme, and lipoprotein lipase levels with reduced impact on hepatic fatty acid synthetase and malic enzyme activity	[103]
**Neuroprotective Activity**					
Eth Ext	Albino rats (neuro-inflammation induced with lipopolysaccharides)	14 days-200 mg/kg and 400 mg/kg	Estimate antioxidant enzymatic levels in brain and neuronal damage	Decreased TBARS level associated with increased neural regeneration; decreased cell edema; increased glutathione; superoxide dismutase; and catalase.	[109]
Alc Ext	Wistar albino rats (injected with 6-hydroxy dopamine)	30 days-200 and 400 mg/kg	Anti-Parkinson’s activity	Reduced oxidative stress, increased dopamine (2.45 ng/mg of protein) and complex I activity that restores locomotor function, and decreased iron asymmetry ratio	[110]
**Radioprotective Activity**					
Hydro-alcoholic stem extract	Gamma radiation-exposed Swiss albino mice	Single dose-200 mg/kg	Cell cycle progression, Spleen colony forming units, micro-nuclei induction	30-day survival rate of 76.30% compared to 100% mortality in control, raised spleen CFU count to 31.60 (treated) compared to control, and decreased induction of micronuclei in the S-phase cell population	[60]
Aqs Ext	Swiss Albino mice (exposed to 60Co gamma radiation)	Single dose for 3, 7, 15 days at a dose of 5, 10, 15 mg/kg	Animal behaviour and survival rate	50% survival rate in 24 days, 100% mortality in 30 days (15 days), 33% survival rate (single dose) for 30 days, 100% mortality in 3 and 7 days, and 50% survival rate in 24 days	[76]
50% hydro-Alc Ext	Swiss albino strain ‘A’ mice (exposure to 60Co gamma radiation)	200 mg/kg	Phagocytic activity, splenocyte proliferation assay, macrophage functionality	19–2% reduction in apoptosis, spleen weight increment, 120% macrophage adherence, splenocytes proliferation	[111]
**Antiallergic Activity**					
Aqs Ext	Albino rats (Bovine albumen and Freund’s adjuvant), Guinea pigs (histamine-induced bronchospasm), Swiss mice (Bovine albumen and Freund’s adjuvant)	100 mg/kg (for 24 h)	Bronchospasm and mast cell production response	95% reduction in bronchoconstriction, capillary permeability, and mast cell number	[112]
Hydroalcoholic extract	Ovalbumin administration i.p. and intranasally in BALB/c mice	7 days-100 mg/kg	Cytokine production, oxidative stress markers	Enhanced superoxide dismutase, glutathione peroxidase, catalase, reductase, reduced airway hyper-responsiveness, IgE and eosinophil level, dwindling level of pro-inflammatory cytokines (COX-2, iNOS, ICAM-1)	[10]
Aqs Ext	Clinical patients (suffering from rhinitis)	60 days-One tablet thrice daily (300 mg extract)	Response to allergic rhinitis	100% nasal discharge control, 83% sneezing relief, decreased eosinophil and neutrophil in nasal smear, nasal mucosa looked pink against blue colour	[113]
**Cardioprotective Activity**					
Isolated *Octacosanol*	Swiss albino mice (injected with Ehrlich ascites tumour cells)	6 to 14 days-60 μg/alternate day	Micro-vessel density, peritoneal angiogenesis, quantification of vaso-endothelial growth factor (VEGF)	Decreased geletinolytic activity on metalloproteinases in dose-dependent manner, inhibition of Nf-κB in VEGF gene expression	[114]
Alc Ext	Myocardial ischemia induced Sprague Dawley rats	7 days-250, 500, and 1000 mg/kg	Testing of infarct size in heart tissue and lipid peroxide levels in liver	Decreased heart rate, a dose-dependent increase in infarct size, and elevated amounts of malonaldehyde in the blood and heart	[115]
Alc Ext	CaCl_2_ induced cardiac arrhythmia in Wistar albino rats	Single dose-150, 250, and 450 mg/kg	Cardiac responses and mineral level in blood	Reduced heart rate, normalized PQRST waves, elevated K levels, lowered Ca and Na levels in the blood, and regulated atrial and ventricular fibrillation	[39]

**Table 4 molecules-28-07073-t004:** Several tentative developed mechanisms are as follows.

S. No	Experimentation/ Animal Model	Targeted Phytoconstituent	Mechanism of Action	References
1	Immunomodulatory activity in murine model	Low molecular wt. phytochemical, polysaccharides as immunostimulatory protein (ImP)	Mitogenic activity on splenocyte and thymocytes Stimulate macrophage phagocyte and bactericidal activity without hemagglutination activity.	[46]
2	Immunostimulatoryactivity in HEK293 cells	α-glucan, (1,4)-a-D-glucan (RR1)	Potent stimulation of Th1 cytokine response in natural killer, T, and B cells RR1 stimulate phagocytic activity of RAW264.7 macrophages independent of CD11b surface expression. RR1, unlike G1-4A, exerting immunostimulatory activity functioning as a TLR6 agonist in HEK293 cells.	[45]
3	Immunomodulatory activity in murine model	β-glucans	β-glucans interact with their cognate receptors on macrophages (like CD11b, TLR2, TLR6, etc.) stimulate a Th1 cytokine response. β-glycosidic linkage found in β-glucans was sine quo non for immune-enhancing activity. Linkages in α-glucans impart immunostimulatory activity.	[119,120,121,122]
4	Examination of B-cells and macrophages as potential target cell populations for G1-4A in murine model	Arabinogalactan polysaccharide G1-4A	Fluorescence microscopy identified B-cells (minor) and macrophages (major) as targets of G1-4A, presumably speculate G1-4A and LPS as the same cellular target. Activity confirmed by anti-TLR4 (toll-like receptor 4) antibodies that demonstrated G1-4A acts as a TLR4 agonist and stimulates murine B-cells which increase lymphocyte proliferation and splenic cellularity. G1-4A activates murine macrophages, by increasing phagocytosis, dependent upon ERK and NF-jB. G1-4A elicited TLR4-MyD88 dependent Th1 cytokine response characterized by upregulation of TNF-α and IL-1β. M1 phenotype highlighted by increased MHC-II and CD-86 surface expression in murine macrophages. Pharmacologic inhibitors demonstrated the role of key cell signaling pathways, including p38, ERK, and JNK MAPKs, in macrophage activation by G1-4A.	[43,44,123]
5	Immunostimulatory studies in Balb/c mice	Glycosides-Cordifoliosides A and B	Immunopotentiation activity by increasing IgG antibody production in Balb/c mice subcutaneously injected with sheep red blood cells.	[35]
6	Immunostimulatory studies in Balb/c mice	Cordifolioside A, syringin, cordiol, cordioside	Increased antibody production. Increased peritoneal macrophage phagocytic activity.	[26]
7	Immunomodulatory activity in murine model	Arabinogalactan polysaccharide	G1-4A in aqueous extract showed polygenic mitogenic activity in B-cells. G1-4A mitigate host immune responses in BALB/c murine model of drug-resistant Mycobacterium tuberculosis.	[23,44]
8	Immunostimulation in murine model of septicemia	G1-4A polysaccharide	Pre-treatment with G1-4A prevented lipopolysaccharide (LPS)-induced mortality in a murine model of septicemia. Reduced mortality was associated with a blunted tumour necrosis factor-alpha (TNF-α) response Increased circulating TNF receptor levels, and decreased nitric oxide release by splenic adherent cells.	[43]
9	Histocompatibility effect in murine model for immunomodulation	Polysaccharide	G1-4A reduce pulmonary bacillary burden, increase Th1 cytokine and decrease Th2 cytokine profile. Pre-treatment of murine RAW264.7 macrophages significantly induce surface expression of major histocompatibility complex-II (MHCII) and CD-86, markers of activated macrophages (M1). M1 macrophages, exhibit microbicidal activity characterized by increased elaboration of pro-inflammatory cytokines and nitric oxide.	[44]
10	Hot water/methanol: water followed by fractionation with n-hexane, ethyl acetate, chloroform, n-butanol, water, ethyl acetate investigating immunomodulatory activity in murine model	N-formylannonain, cordiofolioside A, magnoflorine, tinocordiside, syringin, 11-hydroxymuskatone, N-methyl-2-pyrrolidone, magnoflorine, tinocordioside	Increased neutrophil phagocytosis. Increased phagocytosis and release of reactive oxygen species–biomarkers of enhanced PMN activity. Murine splenocyte proliferation assay showed immunoenhancing effect in aqueous extract.	[30,39]

## Data Availability

Data available on request.

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
