# Peer review of "Unveiling Various Facades of Tinospora cordifolia Stem in Food: Medicinal and Nutraceutical Aspects"

_molecules, 2023, doi:10.3390/molecules28207073_

Round 1
Reviewer 1 Report
The authors studied the phytochemical constituents of Tinospora Cordifolia and their biological activity focusing on medicinal and nutraceutical aspects. The authors reviewed 132 references.
The manuscript is suitable for publication after minor revision.
Title, Abstract line 2: Tinospora Cordifolia should be written in Italic.
Title: „A” is unnecessary.
Abstract line 2: side effect
The first three paragraphs of Introduction are meaningless.
Page 2., paragraph 4. line 1.: universe, the capital letter is unnecessary.
Page 2., paragraph 5. last line: Vitamin, the capital letter is unnecessary.
2. Phytocehmistry of T. Cordifolia, T. Cordifolia should be written in Italic.
Figure 1. The stereochemistry is not given, where it would be interesting, e.g. columbin, pregnane, glycosides, etc. Columbin does not belong to the alkaloids, because it does not contain nitrogen. Columbin C20H22O6 CID 188289 https://pubchem.ncbi.nlm.nih.gov/compound/Columbin#section=Related-Records
Page 2. last paragraph, lines 8, and 12. columbin, instead of columibin
Page 2. last paragraph, lines 12: sitosterol, the capital letter is unnecessary.
2.1 second paragraph: The extract…
Page 15, second paragraph, line first: food, the capital letter is unnecessary.
Figure 3.: the reference is missing.
Figure 4. is readable with difficulty and the reference is missing.
Table 4. should be written in bold and in a separate line.
2.4.1 T. Cordifolia should be written in Italic.
Figure 5. is unreadable and the reference is missing.
Title: ...A Medicinal and Nutraceutical Aspects „A” is unnecessary.
Author Response
Response to Reviewer 1 comments
The authors studied the phytochemical constituents of Tinospora Cordifolia and their biological activity focusing on medicinal and nutraceutical aspects. The authors reviewed 132 references.
The manuscript is suitable for publication after minor revision.
Title, Abstract line 2: Tinospora Cordifolia should be written in Italic.
Answer: We are thankful to reviewer for comments, we made changes according to comments
Title: „A” is unnecessary.
Answer: We are thankful to reviewer for comments, we made changes according to comments
Abstract line 2: side effect
Answer: We are thankful to reviewer for comments, we made changes according to comments
The first three paragraphs of Introduction are meaningless.
Answer: We are thankful to reviewer for comments, we made changes according to comments and removed from introduction part.
Page 2., paragraph 4. line 1.: universe, the capital letter is unnecessary.
Answer: We are thankful to reviewer for comments, we made changes according to comments.
Page 2., paragraph 5. last line: Vitamin, the capital letter is unnecessary.
Answer: We are thankful to reviewer for comments, we made changes according to comments.
- Phytocehmistry of T. Cordifolia, T. Cordifoliashould be written in Italic.
Answer: We are thankful to reviewer for comments, we made changes according to comments.
Figure 1. The stereochemistry is not given, where it would be interesting, e.g. columbin, pregnane, glycosides, etc. Columbin does not belong to the alkaloids, because it does not contain nitrogen. Columbin C20H22O6 CID 188289 https://pubchem.ncbi.nlm.nih.gov/compound/Columbin#section=Related-Records
Answer: We are thankful to reviewer for comments, we made changes according to comments, now figure 1 as shown in below
Page 2. last paragraph, lines 8, and 12. columbin, instead of columibin
Answer: We are thankful to reviewer for comments, we made changes according to comments.
Page 2. last paragraph, lines 12: sitosterol, the capital letter is unnecessary.
Answer: We are thankful to reviewer for comments, we made changes according to comments.
2.1 second paragraph: The extract…
Answer: We are thankful to reviewer for comments, we made changes according to comments.
Page 15, second paragraph, line first: food, the capital letter is unnecessary.
Answer: We are thankful to reviewer for comments, we made changes according to comments.
Figure 3.: the reference is missing.
Answer: We are thankful to reviewer for comments, we created the figure 3, not take from any previous publication.
Figure 4. is readable with difficulty and the reference is missing.
Answer: We are thankful to reviewer for comments, we made changes according to comments, now figure 4 stand as follow
Table 4. should be written in bold and in a separate line.
Answer: We are thankful to reviewer for comments, we made changes according to comments.
2.4.1 T. Cordifolia should be written in Italic.
Answer: We are thankful to reviewer for comments, we made changes according to comments.
Figure 5. is unreadable and the reference is missing.
Answer: We are thankful to reviewer for comments, we made changes according to comments. We revised figure 5 as readable.
Reviewer 2 Report
Presented paper is a review dedicated to Tinospora cordifolia - indigenous to tropical regions plant. According to the authors, this plant is a rich source of diterpenoid lactones, steroids, alkaloids, and glycosides, whereby has a wide range of biological activities and is widely used in Indian medicine. The authors brought together a large number of studies investigating the phytochemistry of T.cordifolia, its use in pharmacology and as food supplements.
On the other hand, I am alarmed by the heavy emphasis on alternative medicine - Ayurveda is often mentioned, no references are given for a number of claims of biological activity, a number of references to questionable journals are given, e.g. An International Quarterly Journal of Research in Ayurveda (Ref 117).
Please check the manuscript carefully for typos, for example:
Page 2 - Each part of the drug i.e leaves, stems, and roots…– maybe, “each part of the plant”?
Page 8 – “various categories likes” – should be “like”
Page 19 – I don’t quite understand this sentence: “Rawson and his co-worker [129] incorporated debittering agent β-cyclodextrin at 3.0% level in ultrasonication proved to reduce bitterness in 10 min further away the tolerance level (12 h) with magnetic stirring and sonicating (30 min), respectively.” Probably “12h” is dedicated to magnetic stirring?
Also, I have a few comments requiring authors’ attention.
Page 1 – “Hippocrates is credited with the 2,500-year-old adage, “Let food be thy medicine and medicine be thy food”- it’s, probably, misquotation. See 10.1016/j.clnme.2013.10.002.
Page 2 – “Across the Universe” - that sounds strange to me, maybe, “across the world”?
Page 2 – “The plant is also rich in carbohydrate, fibers, and proteins. In small concentration Vitamin C, calcium and iron are also present” – need reference.
Page 2 – It seems wrong to me to simply list the dozens of compounds of T.cordifolia. Try rephrasing it by specifying classes of compounds instead of individual components.
Page 2 – I don’t understand how the bitter flavour relates to metabolic activity.
Page 4 – “The nutritional value of …” - need reference. Also, a strange set of characteristics is selected – fat, K, Ca, Fe, Cr. Are they most important? Please, rephrase.
Page 5 - (Sharma et al., 2020) – should be replaced with reference number in square brackets.
Page 14 – “The survey conducted…” - need reference.
Page 15 – “Additionally, it helps prevent chronic diseases” – this sentence badly connects with other text.
Page 15 – What is suitable epidemiological goal for nutraceuticals?
Page 15 – Fig. 3 is describing steps involved in nutraceutical development. This figure doesn’t show how inclusion of nutraceuticals into regular diets may help prevent pathological conditions by delaying or preventing the need for pharmaceuticals using.
Page 16 – “Additionally, it elevated leucocytes counts etc.” – need reference
Page 16 – I think Fig. 4 needs explanatory commentary about declared mechanism of action
Page 20 – “No significant changes were observed on extraction time on concentration of active constituents by adsorption chromatographic fingerprinting at low and high wavelength confirming the existence of some constituents with similar Rf readings in all the prepared samples but in reduced concentration” – Ref 125 doesn’t include chromatographic analysis. Is there another article mentioned here?
Page 22 – “However, the raw form of drug in various states as powder and extract has very unpleasant and bitter, hence the research should mainly concentrate on the T. cordifolia fused product and its expansion do leads to various challenges firstly retaining active phytoconstituents responsible for activity in food product which is essential and require elaborative investigation, secondly the stability of incorporated drug in food products further studied by using various emulsion techniques like double emulsion, nanoemulsion or liposomes” – In my opinion, this sentence too long and hard to understand.
Author Response
We upload the respond to reviewer 2

Round 2
Reviewer 2 Report
The authors have made all the required corrections, however, I've several minor comments:
1. I think that plant name Tinospora cordifolia, as well as it short form (T. cordifolia) should be written in Italic everywhere in the text.
2. Line 64 - "Bernard et al 2018" - please, add reference number in square brackets - [25].
3. Lines 129-131 - "clients than practice assess nutrition higher or inherited aspects of achieving well-being" - probably, it should be: "clients which practice nutrition assess higher or inherited aspects of achieving well-being"?
4. line 279 - What is RSM? To be honest, I couldn't find a definition of this abbreviation in the original source [131].
5. Please check the manuscript carefully for typos, for example:
line 60 - terepenes
line 71 - also reportedthat - missing space
6. Some comments about manuscript structure:
Line 94 - the caption "table 2" should start on the next line
page 5 - table 2 headings are in the middle of the table body
For some reason Fig.1 in the current version of the manuscript looks worse, than in the previous version, also because of decreased resolution.
line 424 - Funding: Funding:
Author Response
Respond to Reviewer 2 comments
The authors have made all the required corrections, however, I've several minor comments:
- I think that plant name Tinospora cordifolia, as well as it short form (T. cordifolia) should be written in Italic everywhere in the text.
Answer: We are thankful to reviewer for comments, we made changes according to comments.
- Line 64 - "Bernard et al 2018" - please, add reference number in square brackets - [25].
Answer: We are thankful to reviewer for comments, we made changes according to comments - Lines 129-131 - "clients than practice assess nutrition higher or inherited aspects of achieving well-being" - probably, it should be: "clients which practice nutrition assess higher or inherited aspects of achieving well-being"?
Answer: We are thankful to reviewer for comments, we made changes according to comments
- line 279 - What is RSM? To be honest, I couldn't find a definition of this abbreviation in the original source [131].
Answer: We are thankful to reviewer for comments, we made changes according to comments
- Please check the manuscript carefully for typos, for example:
line 60 - terepenes
line 71 - also reportedthat - missing space
Answer: We are thankful to reviewer for comments, we made changes according to comments in whole revised manuscript
- Some comments about manuscript structure:
Line 94 - the caption "table 2" should start on the next line
page 5 - table 2 headings are in the middle of the table body
Answer: We are thankful to reviewer for comments, we made changes according to comment
For some reason Fig.1 in the current version of the manuscript looks worse, than in the previous version, also because of decreased resolution.
Answer: We are thankful to reviewer for comments, we made changes according to comments we have change the figure
line 424 - Funding: Funding:
Answer: We are thankful to reviewer for comments, we made changes according to comments
